# Planktonic cnidarian responses to contrasting thermohaline and circulation seasonal scenarios in a tropical western boundary current system

Everton Giachini Tosetto[123], Arnaud Bertrand[1234], Sigrid Neumann-Leitão[3], Alex Costa da Silva[3], Miodeli Nogueira Júnior[5]

[1] MARBEC, Univ Montpellier, CNRS, IFREMER, IRD, Sète, 34200, France
[2] Institut de Recherche pour le Développement, Sète, 34200, France.
[3] Departamento de Oceanografia, Universidade Federal de Pernambuco, Recife, 50670-901, Brazil
[4] Departamento de Pesca e Aquicultura, Universidade Federal Rural de Pernambuco, Recife, 52171-900, Brazil
[5] Departamento de Sistemática e Ecologia, Universidade Federal da Paraíba, João Pessoa, 58051-900, Brazil

*Correspondence to*: Everton Giachini Tosetto (evertontosetto@hotmail.com)

**Abstract.** In western boundary current systems (WBCS), strong currents flow coastward carrying oceanic water masses and their associated planktonic fauna. Variation in the intensity of these currents and in the continental runoff may affect the dynamic interplay between oceanic and coastal communities. In addition, changes in the continental runoff and the thermohaline structure modulate the primary production, adding complexity to the dynamics of these oligotrophic systems. These dynamics likely shape the planktonic cnidarian communities. To further understand such relationships, we used a comprehensive dataset encompassing samples collected above the shelf, slope and around oceanic seamounts and islands of the Fernando de Noronha Ridge in the Western Tropical South Atlantic, in two seasons characterized by distinct thermohaline structure and circulation patterns. Results show that in the tropical South Atlantic and, likely, other western boundary systems with narrow continental shelves, coastward currents spread oceanic waters and their associated cnidarian species over the continental shelf. However, while both costal and oceanic communities co-occur when the continental runoff is notable, oceanic species dominate almost the entire shelf during the dry season characterized by a stronger boundary current intensity. We also conclude that when the mixed-layer depth and associated nutricline is shallower, the enhanced primary productivity supports larger populations of planktonic cnidarian species through a bottom-up control.

## 1 Introduction

Planktonic cnidarians are predators of zooplankton present in most marine environments. Due to their high feeding rates and capacity to reproduce fast and bloom, under specific favorable conditions they have the capacity to control the entire pelagic community with potential impacts on human activities such as fisheries (Boero, 2013; Yilmaz, 2015). Historically, this group was considered a dead-end in the pelagic food web due the low proportion of carbon in their gelatinous tissues. However, recent studies appointed them as a significant food source for many higher trophic level groups, from crustaceans to marine

turtles (Ayala et al., 2018; Hays et al., 2018). These factors, associated to a concern that gelatinous populations could be globally increasing due anthropogenic pressure (Condon et al., 2012; Mills, 2001; Nogueira Júnior et al., 2022; Purcell et al., 2007) enhanced the scientific and popular interest in the dynamics of the group and their role in ecosystem functioning.

The distribution patterns of planktonic cnidarians are closely related to physical processes. Distinct water-masses typically present characteristic species (Pagès, 1992; Pagès et al., 2001; Nogueira Júnior et al., 2014). Oceanic circulation features (e.g. currents, eddies and upwelling/downwelling) transporting these water-masses also convey the associated cnidarian fauna, shaping their meso-scale distribution patterns (Tosetto et al., 2021). Local resource availability and seawater characteristics such as temperature, salinity, and food availability are important factors controlling cnidarian species abundance and distribution, usually at smaller scales (Gibbons and Buecher, 2001; Gili et al., 1988; Luo et al., 2014). Due to these close relationships, among zooplankton communities, planktonic cnidarians are excellent model organisms to understand the complex dynamics between oceanic processes and marine life.

In western boundary current systems (WBCS), strong currents flow coastward carrying oceanic water masses and their associated planktonic fauna over the continental shelf (Dossa et al., 2021; Neumann-Leitão et al., 1999; Thibault-Botha et al., 2004; Tosetto et al., 2021). Under specific conditions (e.g. narrow shelves and low continental drainage), oceanic intrusions may reach coastal areas with typically oceanic species (e.g. *Diphyes bojani*, *Chellophies appendiculata*, *Bassia bassensis*), dominating the entire continental shelf (Tosetto et al., 2021; Thibault-Botha et al., 2004). In the WBCS of the tropical South Atlantic, such conditions were observed in Austral spring, when the continental runoff is reduced and the western boundary current flow is intense (Tosetto et al., 2021). In austral autumn, continental rainfall and temperature are higher and winds are weaker in the area comparing to spring, leading to distinct circulation and thermohaline structure scenarios (Assunção et al., 2020; Dossa et al., 2021). The North Brazilian Undercurrent (NBUC), which flows northwards parallel to the coast and is responsible for the oceanic intrusions over the continental shelf in spring, is weaker and has its core further from the shelf break in autumn (Dossa et al., 2021). Such circumstance, associated to the higher continental discharge, has the potential to reduce the influence of the oceanic species over the continental shelf and favor the spread of coastal species (e.g. *Muggiaea* spp., *Liriope tetraphylla*). However, the effects of the variability in western boundary currents intensity, intrusions of tropical water and river runoff over the continental shelf in zooplankton distribution and abundance are still to be tested.

Offshore, in the Fernando de Noronha Ridge (FNR), a strong stratification occurs. Still, due to wind mixing and the shear between the near-surface westward flowing central South Equatorial Current (cSEC) and the sub-surface eastward South Equatorial Undercurrent (SEUC), the mixed-layer depth is shallower in autumn than in spring (Assunção et al., 2020). A shallower mixed-layer leads to and increase of the nutrient input to the photic layer, enhancing primary production and phytoplankton biomass (Farias et al., 2022). The role of the stratification, mixing and light penetration in the modulation of the primary production is well known (Farias et al., 2022; Mignot et al., 2014; Polovina et al., 1995; Signorini et al., 1999) and it was also related with increased production of zooplankton, though bottom-up control effect (Farstey et al., 2002; Hadfield and Sharples, 1996; Polovina et al., 1995). However, the effects of such changes in zooplankton communities, particularly cnidarians, from highly stratified and oligotrophic tropical systems is still unknown.

Here, we test two main hypotheses. First, an increase in continental runoff and a reduction in western boundary currents intensity would increase the presence of coastal and reduce the presence of oceanic cnidarian species over the continental shelf in WBCS. Second, the shallower the mixed-layer depth, the greater the abundance of planktonic cnidarian communities. For that, we compared the spatial patterns in planktonic cnidarian distribution and abundance from data collected above the shelf, slope and around oceanic seamounts and islands of the FNR in the Western Tropical South Atlantic in spring (Tosetto et al.,

2021) with a new dataset collected in autumn.

## 2 Materials and methods

### 2.1 Data

Data were collected during two "Acoustics along the Brazilian coast" surveys (ABRACOS 1 and ABRACOS 2; Bertrand, 2015, 2017) performed along the Northeast Brazilian continental shelf and slope between 5 and 9°S, and around oceanic

seamounts and islands from FNR, including the Fernando de Noronha Archipelago itself and the Rocas Atoll (Fig. 1). The surveys were carried out on board the French oceanographic vessel R/V ANTEA in austral spring (September-October 2015) and autumn (April-May 2017). Planktonic cnidarians were sorted from zooplankton samples collected at 34 and 45 stations in spring and autumn, respectively (Fig. 1). Samples were collected through oblique hauls, with a Bongo net 300 µm mesh size and 0.6 m mouth opening. The water column was sampled from near bottom to surface over the continental shelf, and from

200 m to the surface in the offshore. The net was towed at approximately 2 knots, at various times of day and night. The net was fitted with a calibrated mechanical flowmeter (Hydro-Bios) to estimate the volume filtered during each haul. Samples were fixed with 4% formaldehyde buffered with sodium tetraborate (0.5 g.l$^{-1}$).

In laboratory, whole zooplankton samples were analyzed under stereomicroscope and cnidarian specimens were identified (mainly according to Pugh, 1999; Bouillon, 1999) and counted. Abundances were standardized in number of individuals per

85 100 m$^{-3}$ for medusae and number of colonies per 100 m$^{-3}$ for siphonophores. For calycophorans, the number of anterior nectophores was used for estimating the polygastric stage abundance, and eudoxid bracts for the eudoxid stage abundance (e.g. Hosia and Båmstedt, 2007; Hosia et al., 2008a). For physonects and Hippopodidae, number of colonies were roughly estimated by dividing the number of nectophores by 10 (Pugh, 1984).

Vertical profiles of temperature (°C), practical salinity and fluorescence were obtained with a CTD-O2 profiler Seabird

SBE911+. Conductivity, temperature and pressure accuracies were estimated at 0.0003 S/m, 0.001 °C and 0.7 dbar, respectively. In addition to CTD-O2, Continuous along-track Sea Surface Practical Salinity data were acquired with a thermosalinograph SBE21. Along-track current profiles were recorded with an 'Ocean Surveyor' ship-mounted acoustic Doppler current profiler (SADCP) operating at a frequency of 75 kHz with a depth range of 15-700 m. SADCP data were processed and edited using the Common Ocean Data Access System (CODAS) software package developed at the University

of Hawaii (http://currents.soest. hawaii.edu). The relative velocities were rotated from the transducer to the Earth reference frame using the ship gyrocompass. The global positioning system (GPS) was used to retrieve the absolute current velocities.

The orientation of the transducer relative to the gyroscopic compass and an amplitude correction factor for the SADCP were determined by standard calibration procedures. Finally, velocity profiles were hourly averaged, providing profiles in the 19-600 m range. SADCP data over the shelf (bathymetry shallower than 70 m) were often contaminated by spurious reflections from the bottom, so the data coverage was only partial in these areas. The current measurements were resampled to 0.1° spatial resolution and depth integrated (0-70 m depth). The analyses of the data obtained by ship-mounted Acoustic Doppler Current Profiler (SADCP) are discussed in Dossa et al. (2021) and Costa da Silva et al. (2021).

## 2.2 Data analysis

For the data analysis, stations were grouped in three systems (Fig. 1): Continental Shelf, WBCS (offshore stations adjacent to coast, mainly over the slope) and South Equatorial Current system (SECS; stations along the FNR). WBCS and SECS present distinct thermohaline structure (Assunção et al., 2020). Stations A2_39 and A2_54 were located in the transition zone between WBCS and SECS and thus removed for the comparative analysis. To dampen effects of dominant species, abundance data was transformed by log (x+1) in all analyses. A Permutational Multivariate Analysis of Variance (PERMANOVA; Anderson et al., 2008) was used to test for diel differences in the community structure of planktonic cnidarians at both seasons. Since no significant differences were observed (Pseudo-F = 0.93842 and 0.86812, P = 0.451 and 0.509, for spring and autumn, respectively) the pooled set of day and night data was used.

Factorial Analysis of Variance (ANOVA) were performed to test for differences in hydromedusae and siphonophores richness and total abundance, and the abundance of the most abundant species (abundance >1%), according to the region (shelf, WBCS and SECS), the season, and the interactions between these factors. When ANOVA was significant, Tukey post-hoc test was used to identify the levels (seasons and regions) that differed among each other.

Spatial patterns in planktonic cnidarian community abundance in autumn were identified by hierarchical cluster analysis using Bray-Curtis similarity matrix. The validity of the groups defined by the cluster analysis was tested through SIMPROF test (5% significance level). A Similarity Percentage (SIMPER) analysis was performed to identify representative species and their contribution to similarity within the groups defined by the cluster analysis. This procedure was previously performed by Tosetto et al. (2021) for the spring dataset.

For each system, constrained ordination analyses were performed to identify associations between the most abundant planktonic cnidarian species and the environmental variables. The following continuous explanatory variables were used: (i) surface layer (0 - 30 m) temperature, (ii) salinity, (iii) dissolved oxygen (except for shelf) and (iv) fluorescence (as an indirect measure of biological productivity), (v) mixed-layer depth, (vi) bottom depth, (vii) zonal component of ADCP data integrated over the first 70 m depth and (viii) meridional component of ADCP data integrated over the first 70 m depth. Seasons (spring and autumn) were used as qualitative explanatory variables. Detrended Canonical Correspondence Analysis (DCCA) revealed a small length of variable gradients (<3), indicating that a linear method was more appropriate to use on this occasion, and thus Redundancy Analysis (RDA) was selected (Lepš and Šmilauer, 2003).

Distribution maps were produced QGIS 3.20 (QGIS Development Team, 2022). Factorial ANOVA was performed in Statistica

10 (StatSoft Inc., 2011). Cluster, SIMPROF, SIMPER and PERMANOVA analysis were performed in Primer v.6 + PERMANOVA (Clarke and Gorley, 2006). DCCA and RDA were performed in CANOCO 4.5 (Lepš and Šmilauer, 2003).

# 3 Results

## 3.1 Environmental background

Whatever the season, surface (0-70 m) SADCP data showed the Central branch of the South Equatorial Current (cSEC) flowing

westward in the open ocean with its core around FNR (Fig. 2a, b). Meanwhile, the NBUC flows northward over the continental slope with intrusions over the continental shelf. The core of both currents meet up in the northwest part of the study area (around 3.5°S, 35°W) forming the North Brazil Current (Bourles et al., 1999; Dossa et al., 2021; Stramma and England, 1999). Although the NBUC was weaker south of 7.5°S in both seasons, it was even weaker in spring 2015 (Fig. 2a) due the presence of an anticyclonic eddy. Thus, intrusions of oceanic waters over the shelf were greatly reduced in the region of the Pernambuco

Plateau in spring (Fig. 2a; Tosetto et al., 2021; Dossa et al., 2021). Such feature was not observed in autumn and thus intrusions over the shelf were observed all along the coast (Fig. 2b). In deeper waters, we observed the core of the NBUC flowing along all the extent of the slope in both season (Fig. 2c, d). However, it was stronger and deeper (upper limit at ~105 m; Fig. 2c), in spring than in autumn, when the core was less intense, shallower (upper limit at ~80 m) and flowing slightly further away from the coast (Fig. 2d). Such differences were reflected in surface currents, which were stronger over the slope in autumn (Fig.

2b), but similar in both season over the continental shelf. (Fig. 2a, b; Dossa et al., 2021). For a detailed description and discussion of circulation patterns and seasonal variability of the NBUC see Dossa et al. (2021).

In both seasons, the WBCS (over the slope) and the SECS (FNR) presented distinct thermohaline structure (Fig. 3b, c, e, f). The thermocline and halocline were sharp in the SECS (Fig. 3c, f), with the South Atlantic Central Water mass (SACW; <13°C) reaching up to ~150 m depth. They were less abrupt in WBCS (Fig. 3b, c) and the SACW was only observed below

250 m (Fig. 3 in Silva et al., 2021; Fig. 3 in Dossa et al., 2021). In both offshore areas, the upper limit of the mixed-layer depth (MLD) was shallower in autumn. However, the difference was much more pronounced in the SECS (90 m and 46 m in spring and autumn, respectively; Fig. 3c,f) than in the WBCS (53 m and 39 m in spring and autumn, respectively; Assunção et al., 2020).

Sea surface temperature was higher in autumn in offshore areas and over the continental shelf (Fig. 3, Fig. A1 in supplementary material).

material). Due the higher temperature, evaporation was intense and sea surface salinity was generally higher during autumn even with the greater rainfall during this season (Fig. 3, Fig. A1 in supplementary material). Specifically, over the continental shelf, distinct patterns were observed in sea surface salinity from thermosalinograph data during each season (Fig. A2 in supplementary material). In spring, as a result of the reduced continental discharge and higher evaporation over shallow areas, surface salinity was  higher in the more inshore areas, with a reduction when approaching the shelf break and offshore.

Differently, in autumn, although the overall salinity was higher, highest values were along the outer shelf/shelf break, with

lower salinity in inshore waters (up to ~30 m isobath) due the higher continental runoff (Fig. A2 in supplementary material). For a detailed description and discussion of the seasonal variability of thermohaline structure, see Assunção et al. (2020).

Over the continental shelf, although presenting high variability, fluorescence was higher in near bottom and in the surface layer in autumn (Fig. 3g), coinciding with period of larger continental discharge and nutrient input. In both the WBCS and the SECS, the peak of florescence reached similar values within the thermocline in both seasons (Fig. 3h, i). However, in the surface layer, while in the WBCS florescence were similar in both seasons (note the small difference at the MLD), in the SECS it was higher in autumn when the MLD and the nutricline were shallower (Fig. 3h, i; Assunção et al., 2020; Silva et al., 2021; Farias et al., 2022).

### 3.2 Species composition

In total, 93 taxa of planktonic cnidarians were sampled in the area, corresponding to 42 hydromedusae, 48 siphonophores, and 3 scyphomedusae. A similar number of taxa were present at each season: 73 in spring and 74 in autumn, with 18 and 19 exclusive species, respectively. In addition, many unidentified larval stages (cerinula, ephyrae and athorybia) were collected (Supplementary table A1; Table 1 in Tosetto et al., 2021). In both seasons, the most frequent and abundant species were almost the same. Among hydromedusae, *Aglaura hemistoma* and *L. tetraphylla* occurred in more than 85% of the samples. However, while *A. hemistoma* was the most abundant in spring, *L. tetraphylla* dominated in autumn. Among siphonophores, *Abylopsis eschscholtzii*, *Abylopsis tetragona*, *B. bassensis*, *C. appendiculata*, *D. bojani*, *Eudoxoides mitra* and *Sulculeolaria chuni* were present in more than 80% of the samples in both seasons and *Eudoxoides spiralis* in spring. *B. bassensis*, *C. appendiculata*, *D. bojani* and *E. mitra* were also the most abundant siphonophores in both cruises, representing 46.5% of the total abundance in spring and 36.9% in autumn (Supplementary table A1; Table 1 in Tosetto et al., 2021).

### 3.3 Spatial and seasonal patterns in diversity and abundance

Hydromedusae species richness was similar all over the study area, with no significant differences among areas or seasons, and averaging $4.5 \pm 2.3$ and $4.1 \pm 1.5$ species per station in spring and autumn, respectively (Table 2, Fig. 4). However, while in spring the total hydromedusae abundance was similar in all areas, averaging $49.9 \pm 66.2$ ind. 100 m$^{-3}$; in autumn it was significantly higher over the continental shelf where it reached up to 1067.6 ind. 100 m$^{-3}$ (383.4 ind. 100 m$^{-3}$ in average; Table 2, Figs. 4, 5; Fig. 2 in Tosetto et al., 2021). No significant seasonal changes were observed in total hydromedusae abundance in the open ocean.

The two dominant hydromedusae, *L. tetraphylla* and *A. hemistoma*, were widespread over the area. *L. tetraphylla* presented higher abundance over the continental shelf in both seasons. However, in autumn it was one order of magnitude higher than in spring in average, reaching up to 770.3 ind. 100 m$^{-3}$ (Table 2, Figs. 5, 6; Fig. 3 in Tosetto et al., 2021). In both seasons, *A. hemistoma* was more evenly distributed than *L. tetraphylla*, with high abundances occurring both over the continental shelf and in the open ocean. But, as occurred with *L. tetraphylla*, it was significantly higher over the continental shelf in autumn (Table 2, Figs. 5, 6; Fig. 3 in Tosetto et al., 2021). The third more abundant hydromedusa taxa, *Aequorea* spp., occurred in low

abundances over the area in spring. However, in autumn its abundance was significantly higher in the SECS, reaching up to 38.3 ind. 100 m$^{-3}$ and was nearly absent over the shelf and in the WBCS (Table 2, Figs. 5, 6).

Siphonophores richness was significantly higher in the two oceanic areas than over the continental shelf, especially in the WBCS where it was also significantly higher in spring (reaching 23 species, $19.5 \pm 2.9$ in average) than in autumn ($13.3 \pm 1.5$ species in average). No differences among seasons were observed over the shelf and in the SECS (Table 2, Fig. 4). The total abundance of siphonophores was similar between seasons over the continental shelf (137.2 and 169.5 ind. 100 m$^{-3}$ in average in spring and autumn, respectively). Meanwhile, a contrasting pattern was observed in the oceanic areas: in spring it was higher

in the WBCS (not statically significant), averaging 153.4 ind. 100 m$^{-3}$, while in autumn significantly higher abundances were observed in the SECS (245.2 ind. 100 m$^{-3}$ in average; Table 2, Fig. 4; Fig. 2 in Tosetto et al., 2021).

Although most dominant siphonophores were widespread over the area, distinct seasonal and spatial patterns were observed. *E. mitra* was more abundant in the SECS in both seasons. However, the pattern was more pronounced in autumn when this species reached up to 109.5 ind. 100 m$^{-3}$. *A. eschscholtzii*, which occurred in similar abundances in the three areas in spring,

was significantly more abundant in the SECS in autumn, reaching up to 67.6 ind. 100 m$^{-3}$. In the WBCS, average abundances of *E. mitra* and *A. eschscholtzii* were similar in spring and autumn (Table 2, Figs. 5, 6; Fig. 4 in Tosetto et al., 2021). Otherwise, over the continental shelf and in the WBCS, *D. bojani* and *B. bassensis* were more abundant in spring than in autumn, while in the SECS the opposite was observed and these species were significantly more abundant in autumn (Table 2, Figs. 5, 6; Fig. 4 in Tosetto et al., 2021).

In spring, *C. appendiculata* was slightly more abundant over the continental shelf and in the WBCS than in the SECS. Although, in autumn its abundance was higher over the continental shelf and lower in the two oceanic areas, seasonal changes were not significant (Table 2, Figs. 5, 6; Fig. 4 in Tosetto et al., 2021). Otherwise, *A. tetragona* and *E. spiralis* were more abundant over the continental shelf and in the WBCS than in the SECS in spring. Conversely, in autumn they were less abundant in the shelf and WBCS than in autumn and the abundances of these species were similar in the three systems (Table

2, Figs. 5, 6; Fig. 4 in Tosetto et al., 2021).

Distinctly from other dominant siphonophores, in both seasons *M. kocchii* occurred almost exclusively over the continental shelf. However, while in spring the species was restricted to few stations in the south of the area, in autumn it occurred in higher abundances and was distributed over most of the continental shelf, except a few stations in the outer shelf (Table 2, Figs. 5, 6; Fig. 4 in Tosetto et al., 2021). No spatial and temporal patterns were identified in the abundance of *S. chuni*.

**3.4 Community structure**

In spring, two main groups of stations were depicted in the Cluster analysis. Group A was represented by four stations over the shelf in the southernmost part of the study area (Fig. 7b). It was mainly characterized by the large abundances of *M. kochii*, *A. hemistoma* and *L. tetraphylla*. It also differed from the other groups by the absence or low abundance of other dominant siphonophores. Group B was represented by the remaining neritic stations and all stations of the WBCS and the SECS where

siphonophores such as *D. bojani*, *B. bassensis*, *A. tetragona*, *C. appendiculata* and *E. mitra* were more abundant. This group

was further subdivided in several subgroups, which differed among each other in the abundance of dominant species. For a detailed description and discussion of the cnidarian community structure in the area in spring, see Tosetto et al. (2021).

Similarly, two main groups were depicted in autumn (Fig. 7c). Group X was composed by shallow stations (bottom depth <25 m) over the continental shelf. SIMPER analysis indicated an average similarity of 63.1%. It was mainly characterized by the occurrences in large or medium abundance of *L. tetraphylla*, *M. kocchii*, *A. hemistoma*, *A. eschscholtzii*, *C. appendiculata* and *B. bassensis*. One outlier was present in this branch (station autumn_3) and it differed from the main branch due the absence of *A. hemistoma*, *A. eschscholtzii*, *C. appendiculata* and *B. bassensis* (Table 3).

Group Y was composed by the deeper stations over the continental shelf and all stations in the WBCS and the SECS. This group was subdivided in two large branches and several outliers. Subgroup Y1 was composed by most stations of WBCS over the slope. SIMPER analysis indicated an average similarity of 73.4% within the group (Table 3). It was represented by the species *C. appendiculata*, *D. bojani*, *E. mitra*, *B. bassensis*, *A. tetragona*, *A. hemistoma*, *L tetraphylla*, *A. eschscholtzii*, *Lensia meteori*, and *E. spiralis*, occurring in high or medium abundances. Subgroup Y2 (average similarity of 78.1%) included most stations in SECS, represented by the species *D. bojani*, *B. bassensis*, *E. mitra*, *A. eschscholtzii*, *A. hemistoma*, *C. appendiculata*, *A. tetragona*, *S. chuni*, *Cordagalma ordinatum* and *Aequorea* spp (Table 3). Subgroup Y2 differed from Y1 due the higher abundance of *A. eschscholtzii*, *B. bassensis*, *Aequorea* spp., *D. bojani*, *A. hemistoma*, *E. mitra* and *C. ordinatum*, and the lower abundance of *L. meteori* and *C. appendiculata*.

**3.5 Species responses to environmental gradients**

The two first canonical axes of the RDA explained 45.9, 28 and 51.4% of species variance for the shelf, SBCS and SECS respectively (Table 4). Monte Carlo test showed that the first and all canonical axes together were significant ($p < 0.05$) for the three analyses. Over the continental shelf, the first axis was related to the cross shelf spatial structure, being positively related fluorescence and negatively related to bottom depth. The second axis was related to seasonal variability being positively related to autumn, with stronger northward surface currents and higher temperature, salinity and fluorescence; and negatively related to spring (Fig. 8a, Table 4). In this system, *L. tetraphylla* and *M. kocchii* were positively related with axis 1, such relation reflecting their higher abundances in shallower and more productive coastal waters during both seasons. Other species were negative related with axis 1, reflecting their higher abundances in the areas over the shelf with influence of oceanic waters. Among these species, *A. eschscholtzii* and *D. dispar* were also positively related to axis 2, reflecting their higher abundance during autumn, while *B. bassensis*, *A. tetragona* and *E. spiralis* were negatively related to axis 2, reflecting their higher abundance in spring (Fig. 8a).

In the WBCS, the first axis was mainly related to the seasonal variability, being positively related to spring, with deeper mixed-layer and weaker westward/presence of eastward surface currents (see Tosetto et al. 2021), and negatively related to autumn, with higher temperature and fluorescence. The second axis was positively related to salinity and negatively related to bottom depth and stronger northward surface currents (Fig. 8b, Table 4). In this system, *C. appendiculata*, *B. bassensis*, *A. tetragona*, *E. Spiralis* and *L. Meteori* were positively related to axis 1, reflecting their higher abundance during spring. *E. mitra*, *D. bojani*,

*D. dispar* and *A. eschscholtzii* were negatively related to axis 1, reflecting their higher abundance in autumn, when shallower mixed-layers and higher fluorescence and temperature were observed. *A. hemistoma*, *L tetraphylla* and *S. chuni* were positively related to axis 2 and shallower waters over the slope (Fig. 8b).

In the SECS, the first axis was mainly related to the seasonal variability being positively related to autumn, with higher temperature and fluorescence, and negatively related to spring, with deeper mixed-layer and weaker westward surface currents. The second axis was positively related to stronger northward surface currents (Fig. 8c, Table 4). In this system, *E. mitra*, *A. hemistoma*, *A. tetragona*, *S.chuni*, *B. bassensis*, *D. bojani*, *C. ordinatum. A. eschscholtzii*, *D. dispar* and *Aequeorea* spp. were positively related to axis 1, reflecting their higher abundance in autumn, when shallower mixed-layers and higher fluorescence and temperature were observed. Meanwhile, *L.tetraphylla*, *E. spiralis*, *C. appendiculata* and *L. meteori* were negatively related to axis 1, reflecting their higher abundance in spring (Fig. 8c).

## 4 Discussion

Here, we describe contrasting seasonal patterns in the planktonic cnidarian community in response to ecological and physical forcing in the Western Tropical South Atlantic. Although the species composition of the cnidarian community was similar in spring and autumn, with almost the same species dominating each specific area, the distribution and abundance of species both over the continental shelf and in the oceanic WBCS and SECS were different. We associate these differences to distinct scenarios in the thermohaline structure, circulation and continental runoff.

In spring, over the continental shelf, intrusions of oceanic water masses caused by the coastward currents characteristic of the WBCS (NBUC and cSEC), reached coastal areas. Most of samples were then dominated by cnidarian species that typically dominate in the open ocean, such as the siphonophores *C. appendiculata*, *D. bojani*, *B. bassensis* and *A. tetragona*. In the Western Tropical South Atlantic, the dominance of such oceanic communities over the continental shelf is not exclusive in cnidarians and has been observed for other zooplanktonic groups such as copepods, chaetognaths, thaliaceans and fish larvae (Neumann-Leitao et al., 2008; Neumann-Leitão et al., 1999; Santana et al., 2020; Schwamborn et al., 1999). Meanwhile, the coastal community (group A), with high abundance of *M. kocchii* and *L. tetraphylla*, and absence or reduced abundance of the siphonophores mentioned above, was restricted to a particular area over the Pernambuco Plateau. In this area, oceanic intrusions were reduced due the presence of an anticyclonic eddy centered at 8.9°S, 34.1°W that locally dampened the NBUC system that favoured the oceanward flow of coastal waters (Tosetto et al., 2021; Dossa et al., 2021).

In autumn, although the average velocity and water transport of the NBUC system were weaker, the core of the current was shallower than in spring, what resulted in stronger surface currents. However, it was also more offshore, thus, over the continental shelf, no remarkable differences were observed in surface currents between spring and autumn (Fig. 2b; Dossa et al., 2021). Based on that, similar to spring, the dominance of oceanic planktonic cnidarian species over the continental shelf would be expected.

However, autumn is also the rainy season in Northeast Brazil. With higher continental runoff, the characteristic coastal water can further spreads over the continental shelf, potentially enhancing the extent to which coastal waters species can reach. The combination of these two antagonist processes (larger continental runoff and advection of oceanic waters) produced an interesting response in the structure of the cnidarian community. The sharp outline between "coastal" and "oceanic like" communities present in spring (group A and subgroup B1) was not observed in autumn. Instead, both communities merged and characteristic species of both environments (e.g. *M. kocchii*, *L. tetraphylla*, *A. eschscholtzii*, *C. appendiculata* and *B. bassensis*) coexisted in most stations over the continental shelf (group X). Exception was the closest station to the coast (station autumn_3), where oceanic species were absent or occurred in low abundance; and stations in the outer shelf, which presented the typical oceanic community and were included at group Y. A typical coastal community spreading along the Northeast Brazilian coast was also observed in other zooplankton groups such as copepods, chaetognaths and appendicularians, with large concentrations of larvae from mangrove organisms as well (Ekau et al., 1999; Neumann-Leitao et al., 2008; Neumann-Leitão et al., 1999; Schwamborn et al., 1999). These coastal communities usually spread from 7 up to ~30 km offshore, sometimes merging with the oceanic community, as we also observed in planktonic cnidarians in autumn. This extension is likely affected by the physical constraints discussed above and by species ecological niche. Thus, our first hypothesis was partially correct: with an increase in continental runoff and a reduction in western boundary currents intensity, the range of coastal cnidarian species were indeed increased. But the reduction in the current intensity was not enough to reduce the presence of oceanic cnidarian species over the continental shelf, thus both communities merged and species co-occurred.

The dynamic interaction between oceanic water mass intrusions and the spread of coastal waters and continental runoff is likely to be the main factor affecting the distribution of planktonic cnidarians over continental shelves in WBCS. In narrow continental shelves, when the continental discharge is reduced and intrusions are strong, oceanic communities may dominate almost the entire shelf. This circumstance was observed in our spring data (Tosetto et al., 2021) and similar patterns were observed in the east coast of Africa (Thibault-Botha et al., 2004). When the continental discharge is enhanced and intrusions are strong, the mixing of both water masses may allow the coexistence of species from both environments (although species niche restrictions may act as well). Such circumstances were observed in our data in autumn and in the continental shelf off Southeast and South Brazil, and off South Africa in some seasons (Nogueira and Oliveira Jr., 1991; Nogueira Júnior et al., 2014; Thibault-Botha et al., 2004). Meanwhile, in WBCS with wider continental shelves and larger river discharge, such as the east coast off China (Yangtze River), typical coastal cnidarian species (mainly represented by *Muggiaea* spp.) may spread and dominate almost the entire shelf. Oceanic species are then restricted to the core of the WBC over the outer shelf and slope (e.g. Xu, 2006, 2009; Xu and Lin, 2006).

To further support this hypothesis, unlike WBCS, in the eastern margin of ocean basins, currents typically flow away to the coast. The oceanward flow opens space over the continental shelf for the spread of coastal waters, which also favors the uplift of deeper water masses (Carr and Kearns, 2003). Thus, contrastingly to the patterns observed in WBCS, coastal cnidarian species such as *Mugiaea* spp., *L tetraphylla* and *Obelia* spp. typically dominate the community inhabiting the entire continental shelves in eastern boundary systems, sometimes also spreading beyond the shelf break, while other siphonophores are absent

or occur in low abundance. Such circumstances were observed in eastern boundary systems from the South Atlantic (e.g. Pagès

and Gili, 1991; Pagès et al., 1991; Pagès and Gili, 1992), South Pacific (e.g. Apablaza and Palma, 2006; Rodríguez and Ruiz, 2019; Palma G and Rosales G, 1995) and North Pacific (e.g. Segura-Puertas et al., 2010; Gamero-Mora et al., 2015). The patterns observed in both margins of ocean basins reinforce the major role of ocean circulation in the distribution of planktonic cnidarians both trough advection of oceanic species over the continental shelf and/or the spread of coastal species to the outer shelf.

Back to the Western Tropical South Atlantic, apart from the spread of coastal cnidarian species, the higher freshwater runoff observed in autumn provides an important supply of nutrients to a continental shelf that is quite oligotrophic during most of the year (Brandini et al., 1997; Castro et al., 2006; Ekau and Knoppers, 1999). These nutrients support greater primary production in the rainy season, noted by the general increase in the fluorescence over the continental shelf in our CTDO data in autumn cruise (Fig. 3g). High primary production may lead to a general increase in the abundance and biomass of organism

though trophic web (bottom-up control) and this is likely behind the increase in the abundance of many dominant cnidarian species, such as *M. kocchii*, *L. tetraphylla*, *A. hemistoma*, *C. appendiculata*, *A. eschscholtzii and S. chuni*, in autumn (compared to spring) over the continental shelf (Figs. 5, 8).

A similar process was observed in the oceanic SECS as well. There, the increase in the supply of nutrients that enhanced primary production in autumn occurred because the thermocline between the oligotrophic Tropical Water and the nutrient rich

South Atlantic Central Water was shallower than in spring (Fig. 3; Assunção et al., 2020; Silva et al., 2021). Thus, more nutrients present in the later reached photic layers, becoming available to phytoplankton (Fig. 3i; Farias et al., 2022). The relation between changes in mixing of water masses and light penetration with primary production in tropical systems is dynamic and in some circumstances, a large mixed-layer favors the supply of nutrients to photic layers (Mignot et al., 2014; Polovina et al., 1995; Signorini et al., 1999). However, the opposite was observed in our area, likely due the high stratification

in the system (Farias et al., 2022). Anywise, the increase in primary production related to changes in mixed-layer depth typically reflects in increased production in the upper trophic chain in zooplankton, reflecting a bottom-up control mixed-layer(Polovina et al., 1995). In the SECS, this process and the consequently increase in food availability likely increase the abundance of most dominant cnidarian species as well (e.g. *Aequorea* spp., *D. bojani*, *B. bassensis*, *E. mitra* and *A. eschscholtzii*). Thus, our results support our second hypothesis that the enhanced primary productivity caused by the shallower

mixed-layer depth in oligotrophic systems supports more abundant planktonic cnidarian communities.

In WBCS, with a thicker thermocline and less pronounced seasonal changes in the depth of mixed-layer (Assunção et al., 2020), such increase did not occur neither in primary production (Fig 3h), nor in the abundance of the species referred above (Fig. 5). Instead, species that were dominant over the continental slope in spring, such as *A. tetragona*, *E. spiralis* and *L. meteori*, were significantly less abundant in autumn (Fig.5). One hypothesis yet to be tested to explain these differences is that

in the spring the organisms were pushed by the currents towards the slope, accumulating there. In autumn, the core of the NBUC was weaker and more diffuse, under these circumstances the accumulation of organisms may have not occurred. Accumulation of organisms associated to circulation and sloping topography is not rare among zooplankton (e.g. Cotté and

Simard, 2005; Hazen et al., 2009; Sourisseau et al., 2006), but was never observed in cnidarians. Further approaches considering stratified samples and/or other methods such active acoustics may help to better understanding the interaction among the behavior of these and other species, oceanic currents and bottom topography in the Western Tropical South Atlantic.

**5 Conclusions**

We observed contrasting seasonal patterns in the planktonic cnidarian community in response to ecological and physical processes associated to changes in the thermohaline structure and circulation in the western boundary system of tropical South Atlantic. We concluded that in the tropical South Atlantic and, likely, other western boundary systems with narrow continental shelves, coastward currents spread oceanic waters and its associated cnidarian species, such as *C. appendiculata*, *D. bojani*, *B. bassensis* and *A. tetragona*, over the continental shelf. However, when the intensity of western boundary currents is weaker and under higher continental runoff (as observed in autumn in this study), costal (e.g. *M. kocchii* and *L. tetraphylla*) and oceanic communities merge and co-occur, while in dry seasons (herein in spring), oceanic species may dominate the entire shelf. We also conclude that the enhanced primary productivity in oligotrophic systems caused by the seasonal changes in the depth of mixed-layer that uplift nutrient rich water masses to photic layers, as we observed in autumn, supports larger populations of planktonic cnidarian species through bottom-up control.

**Author contribution:** Conceptualization: AB, SNL; taxonomy: EGT, MNJ; data analysis: EGT; writing - original draft preparation: EGT; writing - contributions, review and editing: all authors. All authors have read and agreed to the published version of the manuscript.

**Data availability:** The authors confirm that the data supporting the findings of this study are available within the article and its supplementary materials.

**Competing interests:** The authors declare they have no conflict of interests.

**Acknowledgements:** We are grateful to the French oceanographic fleet for funding the survey ABRAÇOS 1 and the officers, crew and scientific team of the R/V Antea for their contribution to the success of the operations. The present study was not possible without the support of all members from LABZOO and other laboratories from UFPE and UFRPE. We thank to CAPES (Coordenação de Aperfeiçoamento de Pessoal de Nível Superior) and CNPq (Brazilian National Council for Scientific and Technological Development), which provided Research Scholarships to E.G.T., B.B.S. and S.N.L. This work is a contribution to the LMI TAPIOCA (www.tapioca.ird.fr), CAPES/COFECUB program (88881.142689/2017-01), the European Union's Horizon 2020 projects PADDLE (grant agreement No. 73427) and TRIATLAS (grant agreement No. 817578).

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

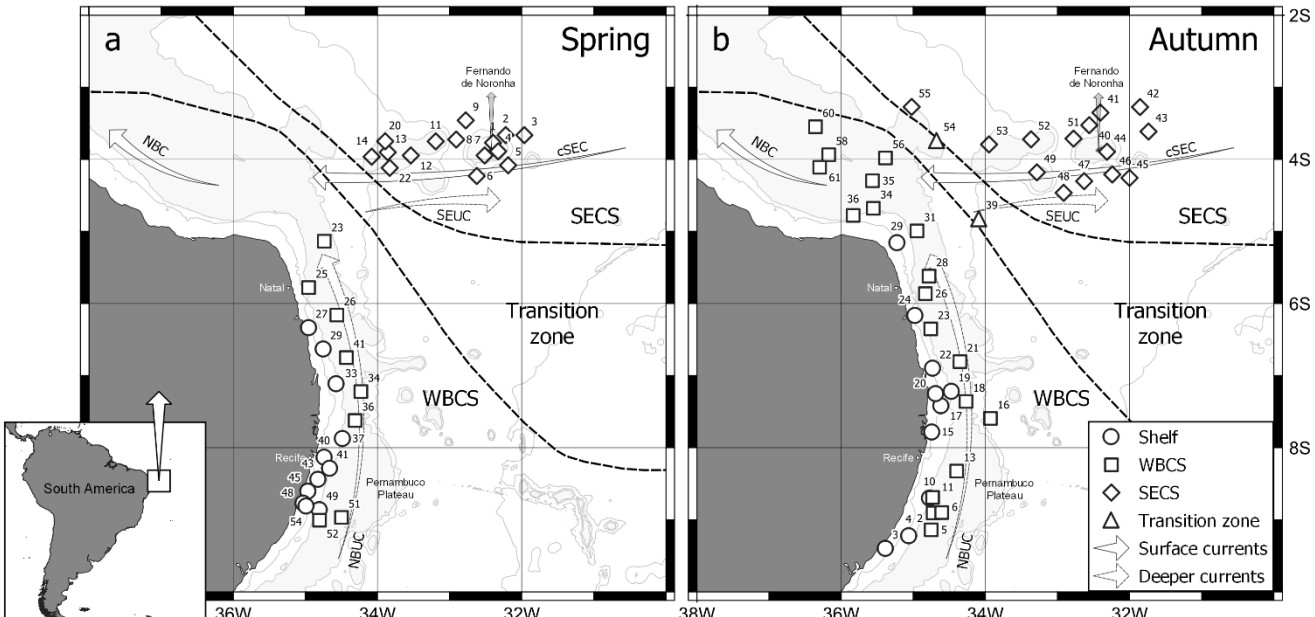

**Figure 1: Geographic location of the study area in the Western Tropical South Atlantic, showing the sampled stations during (a) spring 2015 and (b) autumn 2017.**

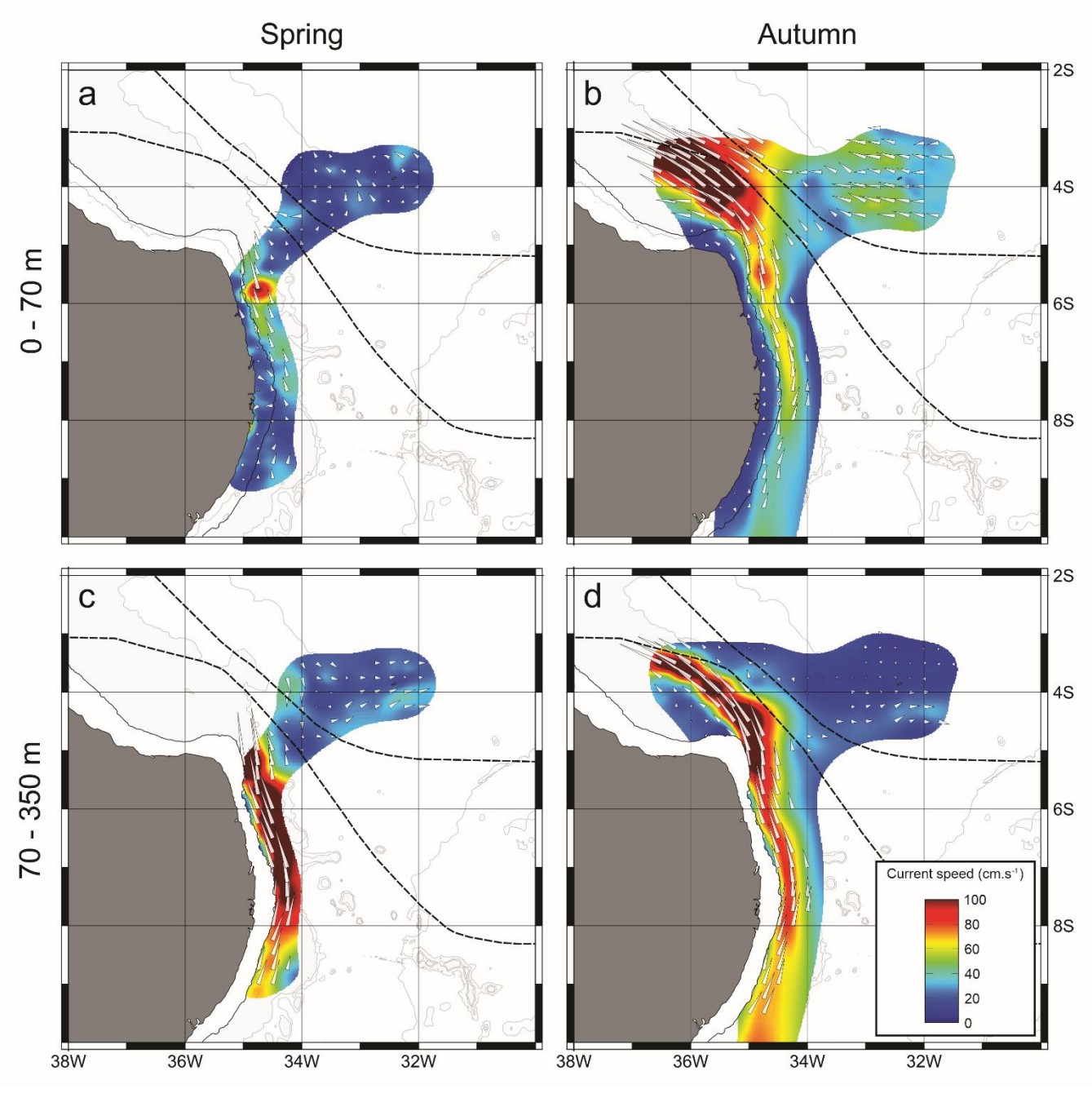

**Figure 2: Surface (0-70 m; a, b) and deeper (70-350 m; c, d) current vectors and velocity of ADCP data during spring 2015 (a, c) and autumn 2017(b, d). Dashed lines indicate boundary between the WBCS, transition zone and SECS.**

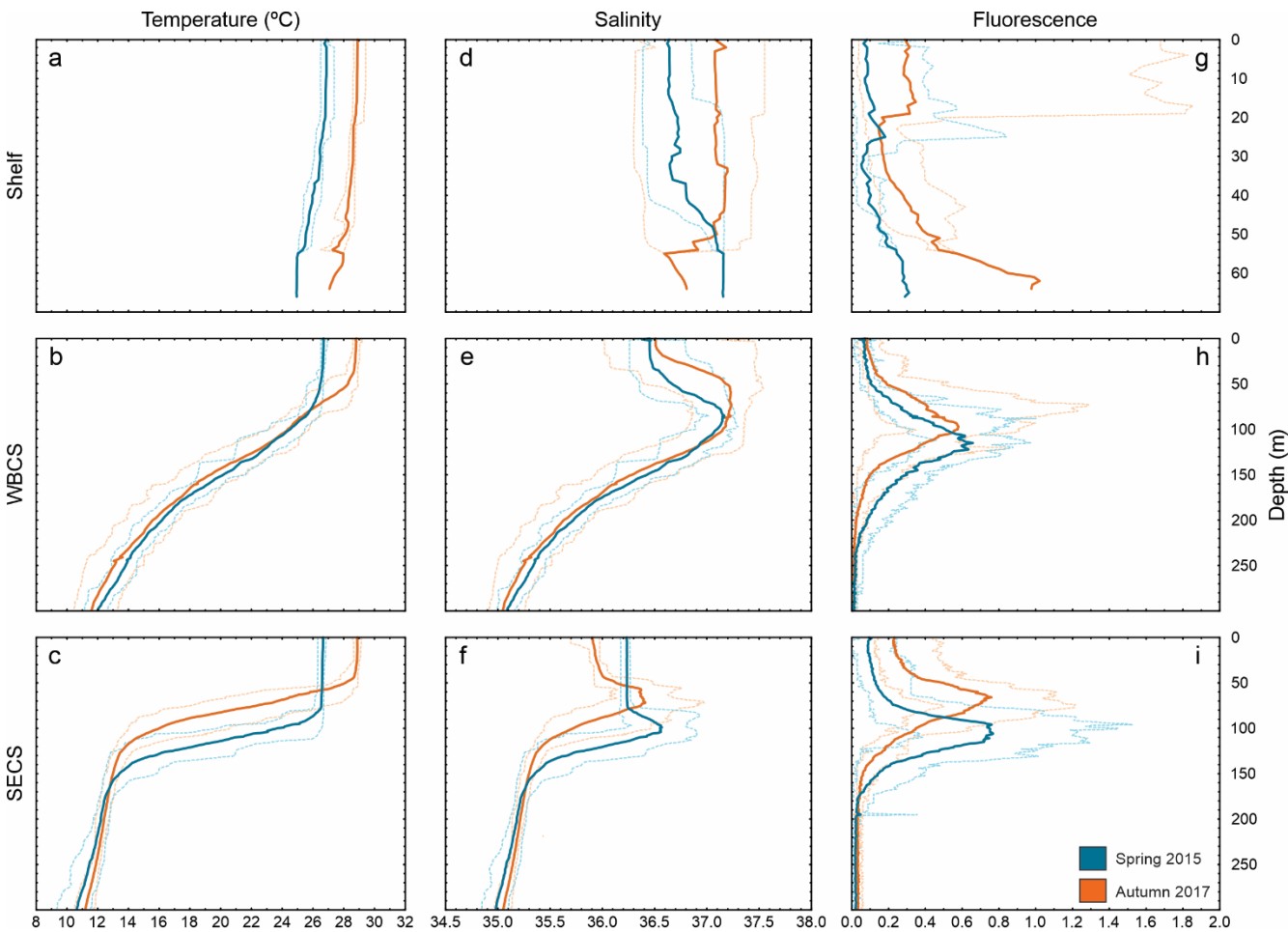

**Figure 3: Vertical profiles of temperature (a, b, c), salinity (d, e, f) and fluorescence (g, h, i) during spring 2015 (blue) and autumn 2017 (orange) in three distinct areas from the Western Tropical South Atlantic. Solid lines are the average among stations and dashed lines are the maximum and minimum, in each season/area.**

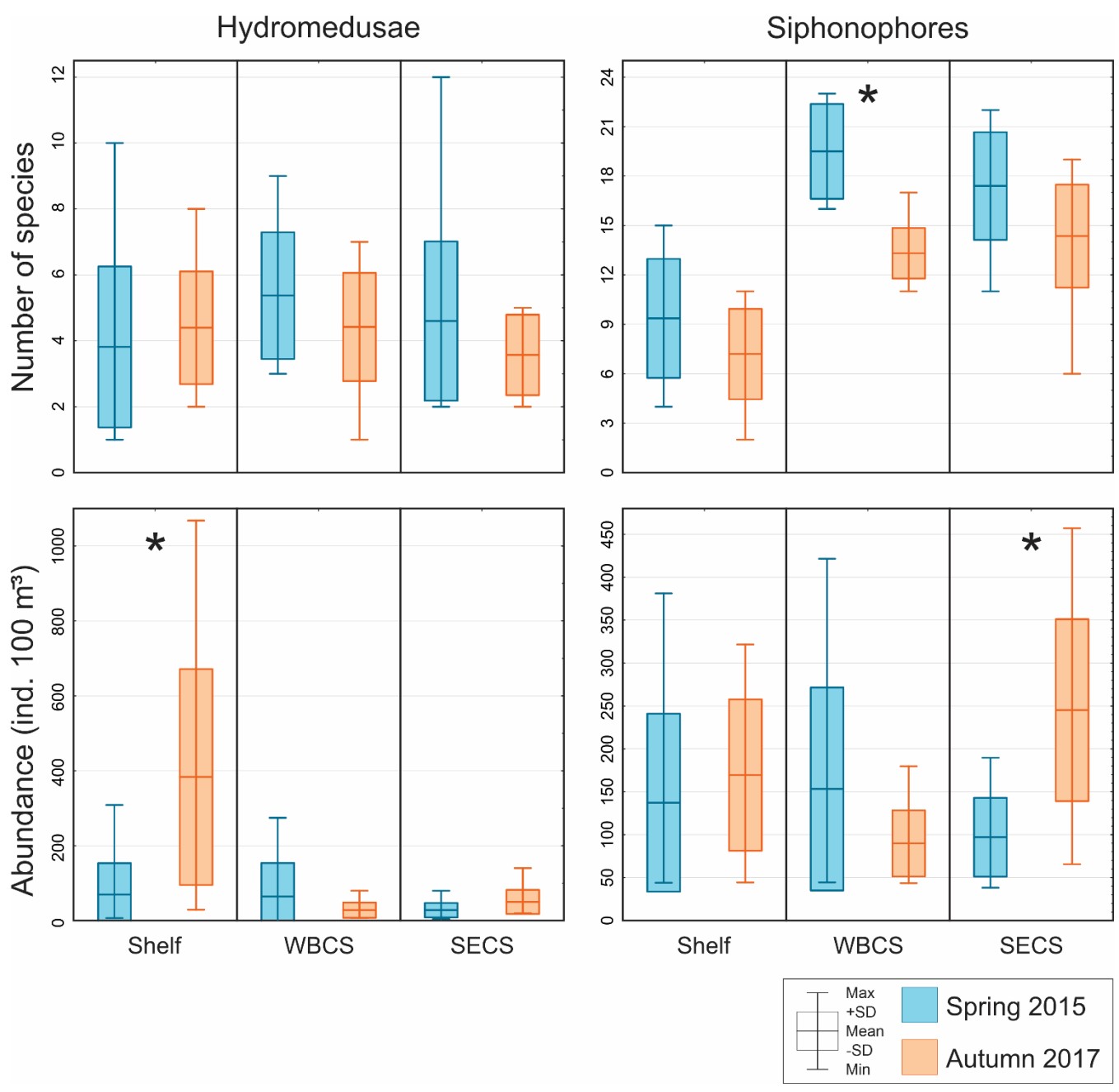

**Figure 4: Comparison of average number of species and total abundance of hydromedusae and siphonophores during spring 2015 (blue) and autumn 2017 (orange) in three distinct areas from the Western Tropical South Atlantic. * Significant seasonal changes in ANOVA.**

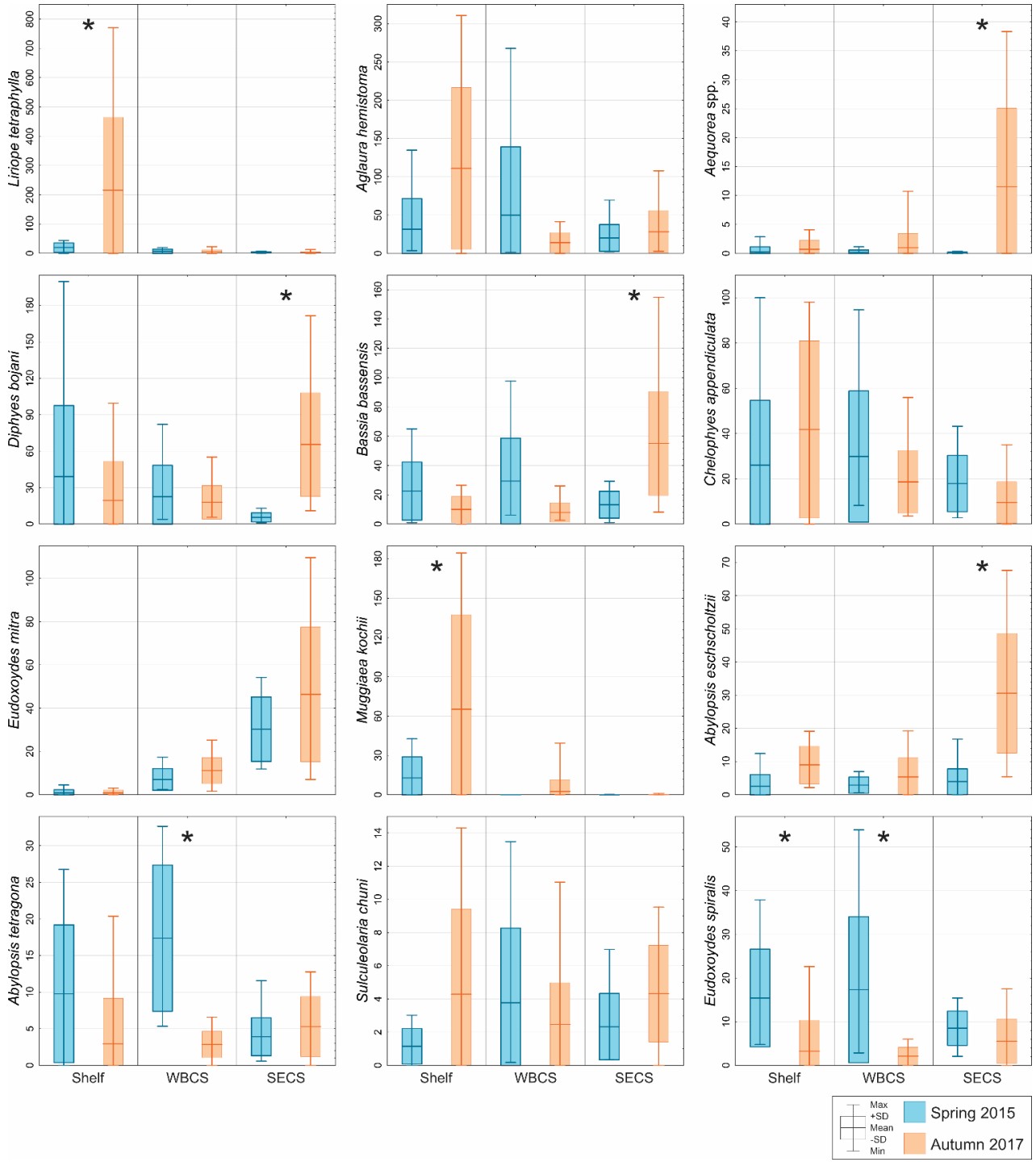

**Figure 5: Comparison of average abundance (ind. 100 m⁻³) of representative species of hydromedusae and siphonophores during spring 2015 (blue) and autumn 2017 (orange) in three distinct areas from the Western Tropical South Atlantic. * Significant seasonal changes in ANOVA.**

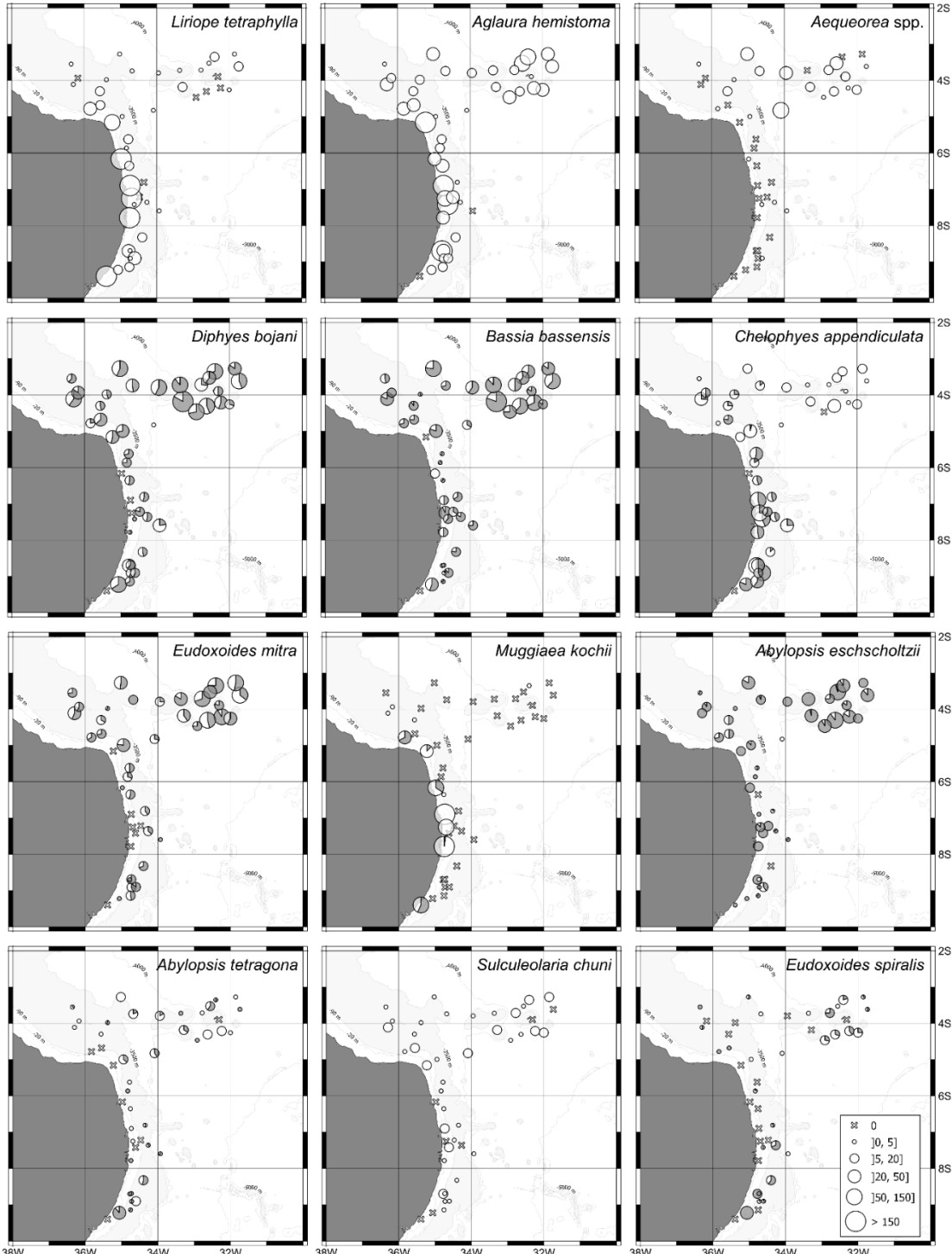

Figure 6: Geographic distribution of representative species of hydromedusae and siphonophores in the first 200 m of the water column in Autumn 2017. For calycophoran siphonophores, black represents polygastric stage and white eudoxid stage.

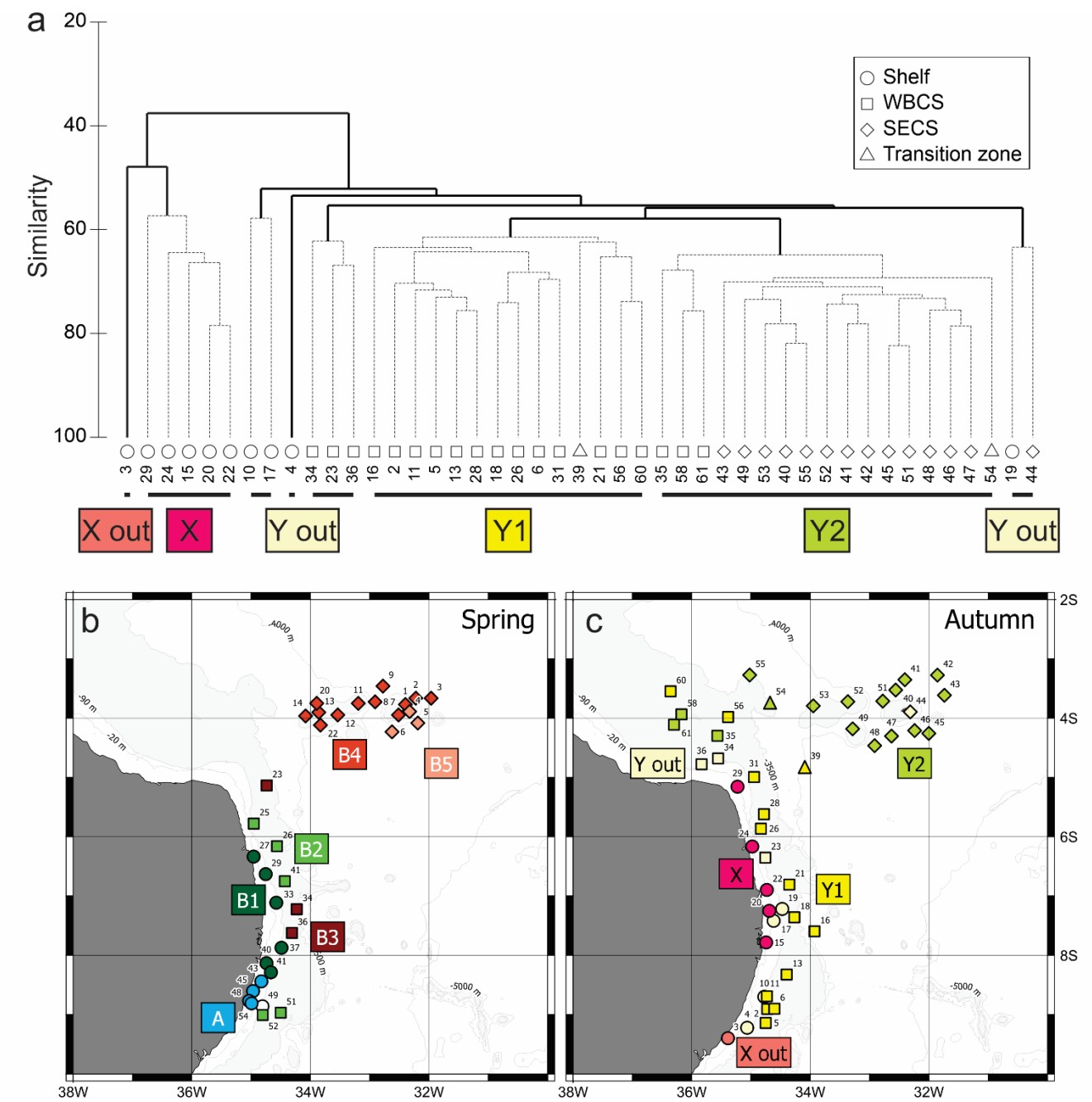

**Figure 7: (a) Cluster analysis dendrogram of autumn 2017 data indicating two main groups and subgroups of stations with similar planktonic cnidarian communities in the Western Tropical South Atlantic Ocean, dashed lines are significant groups in the SIMPROF analysis. (b) Map indicating location of the groups and subgroups arranged in the cluster analysis during spring 2015 (Tosetto et al., 2021). (c) Map indicating location of the groups and subgroups arranged in the cluster analysis during autumn 2017. Dendogram of spring data in Tosetto et al. (2021).**

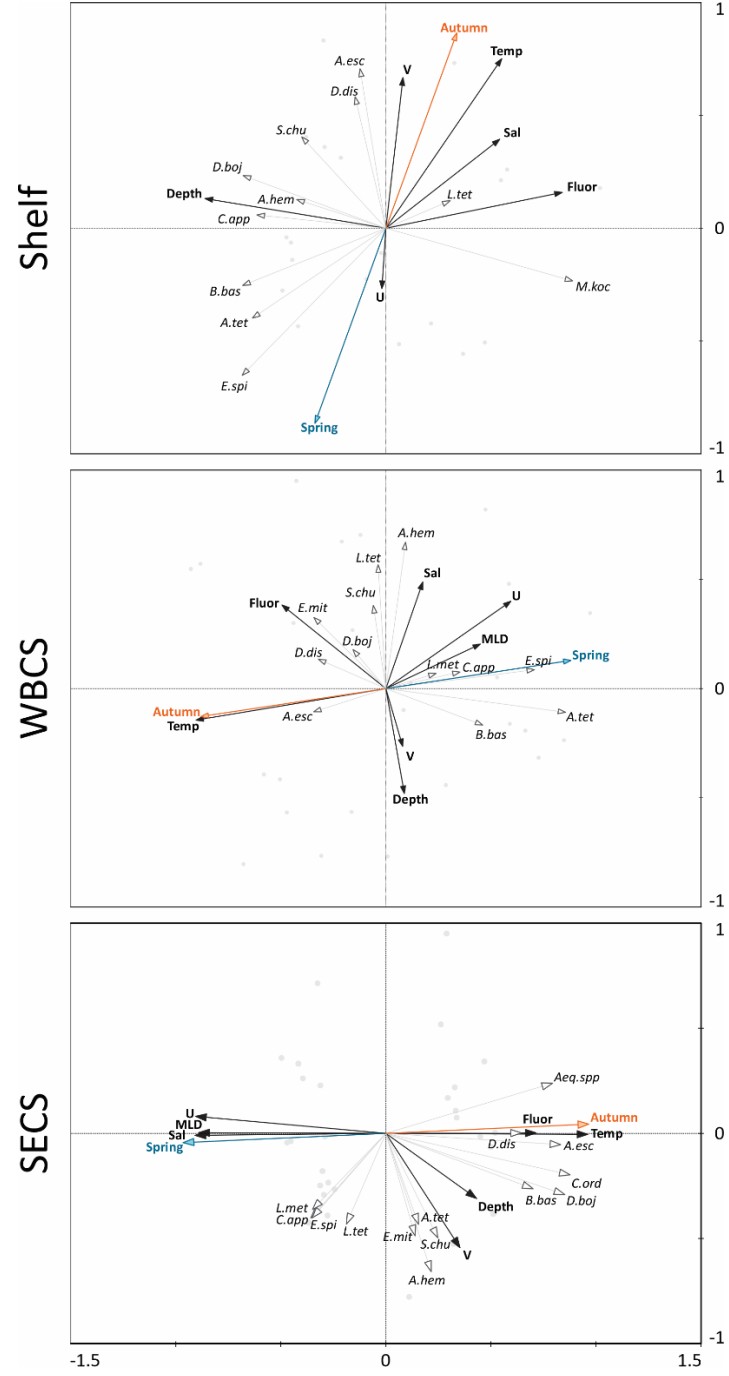

**Figure 8: Redundancy Analyses (RDA) performed between the cnidarian community and environmental explanatory variables from the Continental Shelf, the Western Boundary Current System (WBCS) and South Equatorial Current System (SECS) in the Western Tropical South Atlantic**

**Table 1: List of acronyms**

| Abbreviation | Definition |
| --- | --- |
| cSEC | central South Equatorial Current |
| FNR | Fernando de Noronha Ridge |
| NBUC | North Brazilian Undercurrent |
| SECS | South equatorial current system |
| SEUC | South Equatorial Undercurrent |
| WBCS | Western boundary current system |

**Table 2: Results of the factorial analysis of variance testing differences in community indicators and abundance of representative planktonic cnidarian species among seasons (spring and autumn) and areas (shelf, WBCS and SECS). Significant P-values are in bold.**

| Indicator/species | Seasons | | Areas | | Seasons x Area | |
|---|---|---|---|---|---|---|
| | F | p | F | p | F | p |
| Hydromedusae richness | 1.05 | 0.31 | 1.36 | 0.26 | 1.26 | 0.29 |
| Total hydromedusae abundance | 13.96 | **0.00** | 19.98 | **0.00** | 15.20 | **0.00** |
| Siphonophores richness | 31.62 | **0.00** | 56.56 | **0.00** | 3.10 | **0.05** |
| Total siphonophores abundance | 4.01 | 0.05 | 2.32 | 0.11 | 10.67 | **0.00** |
| *Liriope tetraphylla* | 9.49 | **0.00** | 12.04 | **0.00** | 9.05 | **0.00** |
| *Aglaura hemistoma* | 1.97 | 0.17 | 5.46 | **0.01** | 6.87 | **0.00** |
| *Aequorea* spp. | 8.94 | **0.00** | 6.79 | **0.00** | 7.12 | **0.00** |
| *Diphyes bojani* | 2.36 | 0.13 | 1.38 | 0.26 | 10.81 | **0.00** |
| *Bassia bassensis* | 0.30 | 0.59 | 5.95 | **0.00** | 18.77 | **0.00** |
| *Chelophyes appendiculata* | 0.06 | 0.81 | 5.17 | **0.01** | 2.49 | **0.09** |
| *Eudoxoides mitra* | 3.43 | **0.07** | 42.85 | **0.00** | 1.92 | 0.15 |
| *Muggiaea kochii* | 8.34 | **0.01** | 15.40 | **0.00** | 6.84 | **0.00** |
| *Abylopsis eschscholtzii* | 32.31 | **0.00** | 17.32 | **0.00** | 14.19 | **0.00** |
| *Abylopsis tetragona* | 24.36 | **0.00** | 6.03 | **0.00** | 12.46 | **0.00** |
| *Sulculaeolaria chuni* | 3.04 | 0.09 | 0.24 | 0.79 | 3.14 | 0.05 |
| *Eudoxoides spiralis* | 30.20 | **0.00** | 0.93 | 0.40 | 4.39 | **0.02** |

**Table 3: Results of SIMPER analysis, showing the relative contribution of planktonic cnidarian species in the formation of the groups defined in the Cluster analysis during autumn 2017.**

| Species | X | Y1 | Y2 |
|---|---|---|---|
| *Abylopsis eschscholtzii* | 11.2 | 5.6 | 11.6 |
| *Aequorea* spp. | - | - | 3.7 |
| *Abylopsis tetragona* | - | 8.3 | 5.6 |
| *Aglaura hemistoma* | 19.8 | 7.9 | 11.2 |
| *Bassia bassensis* | 6.2 | 11.2 | 13.6 |
| *Chelophyes appendiculata* | 7.8 | 15.2 | 7.3 |
| *Cordagagalma ordinatum* | - | - | 3.9 |
| *Diphyes bojani* | - | 14.5 | 15.3 |
| *Eudoxoides mitra* | - | 13.7 | 12.4 |
| *Eudoxoides spiralis* | - | 1.1 | 3.6 |
| *Lensia meteori* | - | 4.9 | - |
| *Liriope tetraphylla* | 26.1 | 5.8 | - |
| *Muggiaea kochii* | 21 | - | - |
| *Sulculeolaria chuni* | - | - | 5 |

580

**Table 4: Summary of the Redundancy Analyses (RDA) performed between the cnidarian community and environmental explanatory variables from the Continental Shelf, the Western Boundary Current System (WBCS) and South Equatorial Current System (SECS) in the Western Tropical South Atlantic.**

| | Shelf | | WBCS | | SECS | |
|---|---|---|---|---|---|---|
| | Axis 1 | Axis 2 | Axis 1 | Axis 2 | Axis 1 | Axis 2 |
| Eigenvalues | 0.327 | 0.132 | 0.179 | 0.101 | 0.411 | 0.103 |
| Species-environment correlations | 0.843 | 0.908 | 0.904 | 0.759 | 0.958 | 0.796 |
| **Cumulative variance (%):** | | | | | | |
| Of species data | 32.7 | 45.9 | 17.9 | 28 | 41.1 | 51.4 |
| Of species-environment relationships | 56 | 78.5 | 35.8 | 56.1 | 65.4 | 81.7 |
| **Correlations of explanatory variables:** | | | | | | |
| Spring | -0.3144 | -0.865 | 0.8473 | 0.1287 | -0.9647 | -0.0439 |
| Autumn | 0.3144 | 0.865 | -0.8473 | -0.1287 | 0.9647 | 0.0439 |
| Bottom depth | -0.8021 | 0.1315 | 0.0838 | -0.4804 | 0.4311 | -0.31 |
| Temperature | 0.5136 | 0.7517 | -0.8698 | -0.1441 | 0.9605 | -0.005 |
| Salinity | 0.5049 | 0.3943 | 0.1684 | 0.4868 | -0.9079 | -0.0097 |
| Fluorescence | 0.7822 | 0.1583 | -0.4763 | 0.382 | 0.7149 | 0.0018 |
| Mixed-layer depth | - | - | 0.4322 | 0.2034 | -0.8906 | 0.0029 |
| Zonal currents | -0.0185 | -0.2676 | 0.5719 | 0.3988 | -0.9061 | 0.0805 |
| Meridional currents | 0.0756 | 0.667 | 0.0767 | -0.2647 | 0.3528 | -0.5446 |

585