# Peer review of "Planktonic cnidarian responses to contrasting thermohaline and circulation seasonal scenarios in a tropical western boundary current system"

_EGUsphere, 2022_

## Author Response (AR1)

Dear Prof. Dr. Mario Hoppema,

Please find the corrected MS of our study "Planktonic cnidarian responses to contrasting thermohaline and circulation seasonal scenarios in a tropical western boundary current system".

We sincerely appreciate the useful insights provided by the referees to improve our study. As detailed in the rebuttal letter below (responses in bold) we took into account all corrections and suggestions to prepare the revised version of our MS. The lines mentioned below refer to the new corrected version.

We hope that you will find that the revised manuscript adequately addressed all comments and that it now is suitable for publication in Ocean Science.

We thank you very much, in advance, for the attention you will grant to our re-submission.

**Reviewer 1.**

General comments:

This study is a follow up of Tosetto et al. (2021), in which the authors present the pelagic cnidarian distribution obtained in the ABRACOS 1 survey. In this new manuscript, the authors compare their previously published data (austral spring) with the new, unpublished data from ABRACOS 2 (austral autumn). In this instance, and different from Tosetto et al. (2021), they include water column structure and surface current velocities to address three hypothesis: (1): increases in continental runoff and (2) reduction of western boundary current intensity increase the presence of oceanic cnidarian species over the shelf and (3) a decrease in mixed layer depth is associated to high abundances of planktonic cnidarians. Overall, the paper is very interesting as it shows how pelagic cnidarian communities are closely linked to different water masses and follow the distinction between the west Brazilian continental shelf, the transition zone, and the south equatorial current system, however, more appropriate data analysis to formally address the hypothesis mentioned above will strengthen the manuscript.

For hypothesis 1: can the presence of continental runoff in each sampling station be inferred from surface salinity values or other physical variables? In figure 3, panel d, one can see the large variability in salinity values during autumn compared to spring. I think the authors have some valuable information for each station that is being masked by just presenting the average.

**R.: Thanks for your comment. In the continental shelf off Brazil, due the high temperature and evaporation, salinity is higher than offshore even close (~1 km) to the mouth of estuaries. Thus, at the scale of our CTD samples, the variability we observed in SSS was likely mostly due to diel changes in temperature and evaporation. It was therefore not possible to infer the presence of river runoff from CTD salinity. To deal with that, in the new version, we included**

**along-track thermosalinograph salinity data for the two cruises. Although the overall salinity was lower in spring, as a result of the reduced continental discharge and higher evaporation over shallow areas, we can observe that salinity peaks in the more inshore areas in this season. Differently, in autumn, although the overall salinity was higher (due higher temperature and evaporation), highest values were along the outer shelf/shelf break, with a reduction inshore, matching with the distribution of our cluster groups. Such results and discussion were included in lines 154-160**

For hypothesis 2: I am far from being an expert on regional circulation patterns in the South Atlantic. The comparison between regions (Shelf, WBCS, transition zone and SECS) seems reasonable, and the discussion of the NBUC advecting oceanic plankton into the shelf during spring sounds reasonable. However, given my lack of familiarity with the system, just by looking at the current velocity vectors in Figure 2 (which are from 0-70 m), I infer that the northward surface currents are stronger during fall. Also, the NBUC influences processes from 50-300 m (Veleda et al. 2012), and Dossa et al. (2021) state that the NBUC core is shallower during the fall and thus they observe faster surface currents during the fall (page 10, second paragraph of the discussion). In summary, given the large depth range at which these plankton samples are integrated, it seems important to better correlate the current velocity data with the plankton abundance data (providing more information at deeper depths, differentiating between on-shelf and off-shelf current velocities, etc...) to better support the argument that circulation patterns are the drivers of cnidarian plankton community composition

**R. Although the core of NBUC was stronger in spring, it was shallower in autumn (also demonstrated in Dossa et al., 2021, in Fig. 6). Thus, surface currents were indeed slightly stronger in autumn. However, during this season the core was also further away from the coast and no remarkable differences were observed between seasons over the continental shelf. Our conclusion is that oceanic waters and its associated fauna were advected over the continental shelf at both seasons. We included in figure 2 adcp data for the 70-350 strata, and improved the Discussion in lines 133-145.**

For Hypothesis 3: with CTD profiles for each station, it is possible to infer the mixed layer depth and perform a simple correlation between this property and total cnidarian abundance per station? There might be other variables that could be considered as predictor variables of cnidarian abundance (i.e. chlorophyll concentration). I think this approach will better address your third hypothesis.

**R. Thanks for your suggestion, we included gradient analyses (RDA) for the three systems.**

The data analysis approach presented in Figure 5, 6 and 7 is very appropriate to address hypothesis that deal with differential changes in abundance, distribution and community composition of pelagic cnidarians across seasons and between regions, thus an hypothesis should be made accordingly.

**R. We believe your suggestion was already addressed properly within the mixed-layer hypothesis.**

Grammar corrections:  The ones listed below are only some of the grammar errors that I found throughout the manuscript. I have found it very useful to ask a colleague who is a native English speaker to proofread the manuscript and find other grammar errors. Not being a native English speaker myself, it would be good to have someone else's opinion on this topic:

**R. Thanks for your concern, language was improved in the new version.**

Line 25 sentence two: 'Due **to** their high…' I would recommend rewriting that sentence.

**R. We improved it.**

Line 25 sentence 5: 'recent studies appointed **them…**' I also recommend improving the clarity of this sentence

**R. We improved it.**

Line 230: 'Most of **the** samples..'

**R. Corrected.**

**Reviewer 2.**

Tosetto et al. present a study on the influence of ocean currents and freshwater runoff on cnidarian population. The study is based on an extensive dataset obtained during two long-distance cruises. There is a lack of such extensive datasets and the authors relate the observed cnidarian community structure to the observed physical parameters. I find the study very interesting and I am sure many researchers with an interest in jellyfish will as well. However, I find the manuscript difficult to read and I would suggest several improvements before publication.

I am not a native English speaker, so my comments regarding the language should be taken only as suggestions. However, I find many sentences too lengthy and confusing. I urge the authors to consult a native speaker on this topic. There are numerous abbreviations in the text and a list of acronyms is a must.

**R. Thanks for your concern, language was improved in the new version and a list of acronyms was included (Table 1).**

As many of the potential readers, I am coming from a different part of the World and I am unfamiliar with the hydrography of Brazilian shelf and with many of the observed cnidarians as well. I'd suggest the authors to include a schematic representation of the currents in the study area and refer to it in the introduction and in the section 3.1.

**R. Thanks for the suggestion. We included the schematic representation of currents in Fig. 1.**

Also stating more clearly in the first half of the paper, which species are considered oceanic and which coastal, would make the text easier to read for those who are less familiar with the cnidarians of the South Atlantic.

**R. Thanks for the suggestion we included examples of coastal and oceanic species.**

I think both hypotheses could be better supported by the collected data. E.g. showing that group X could be matched with a lower salinity, would support the hypothesis that stronger continental runoff supports coastal species. Also a correlation between the fluorescence, mixed layer depth and abundance of cnidarians, would be a stronger indicator that the enhanced primary productivity is the cause for cnidarian proliferation. However, the first hypothesis is relatively well supported by Fig. 7. On the other hand, although the second hypothesis seems plausible, the presented support for it is rather weak.

**R. Thanks for your suggestion, we included gradient analyses (RDA) for the three systems.**

I am also surprised that the authors didn't use the satellite data to support their hypotheses. Geostrophic currents based on the the altimetry data could be obtained for the entire area (e.g. from the Copernicus portal: https://marine.copernicus.eu/). Same goes for the surface chlorophyll concentration. I am not insisting the authors should add this, but I am quite sure it would significantly improve the manuscript.

**R. Thanks for the suggestion. Dossa et al. 2021, already compared the same ADCP datasets with geostrophic currents.**

Comments, suggestions and corrections by lines:

L45: "such circumstance was observed in spring" – "such conditions were observed in austral spring"

**R. Thanks for the suggestion. We changed the sentence.**

L54: remove "still"

**R. Thanks for the suggestion. We removed it.**

L55: cSEC eastward? Isn't it westward? And I guess SEUC should be eastward then?

**R. Thanks for the noticing. We corrected both.**

L63: Remove "In this context,"

**R. Removed.**

L75: "sorted" – "identified"

**R. Thanks for the suggestion, however we believed that "sorted" is better in the context of the sentence.**

L88: "10-3" – "10â• »³"

**R. Corrected.**

L98: GPS was used to retrieve velocities or velocity position? Maybe both? Is GPS accurate enough to provide reliable ship velocity in the order of the ocean current speed?

**R. SADCP data were processed and edited using the Common Ocean Data Access System (CODAS) software package developed at the University of Hawaii (http://currents.soest.hawaii.edu). The relative velocities were rotated from the transducer to the Earth reference frame using the ship gyrocompass. The global positioning system (GPS) was used to retrieve the absolute current velocities. The orientation of the transducer relative to the gyroscopic compass and an amplitude correction factor for the ADCP were determined by standard calibration procedures (Joyce, 1989; Pollard and Read, 1989). See also in Dossa et al. (2021).**

L95: "hourly averaged"; remove "located"

**R. Corrected.**

L96: "affected" – "contaminated"; "reflections from the bottom"

**R. Corrected.**

L97: remove "shallow". I'd suggest to rephrase the sentence to something like: "The current measurements were resampled to 0.1° spatial resolution and depth intergrated."

**R. Corrected.**

L101: "grouped into three geographic areas"

**R. Thanks for the suggestion. However, due the distinct current systems in place in the study area, we prefer interpreting them as systems.**

L104-105: "… dominant species, logarithmic abundance was used in all analyses."

**R. Thanks for the suggestion. However, we prefer referring as data transformation, in accordance with statistics literature.**

L129: "intrusions over the shelf" – Of what? Be more specific. Oceanic water probably.

**R. We changed to "intrusions of oceanic waters over the shelf".**

"Pernambuco Plateau" – What is this? Should be marked in Fig. 1

**R. Thanks for the suggestion. We marked it in Fig. 1.**

L130-131: "intrusions over the shelf were observed" – how were they observed?

**R. Transport of water was observed by the coastward currents.**

L 134: Regarding the fact that the circulation is an important part of the study, I don't think it is enough to reference a paper. As said, a schematic would help. And a better description as well.

**R. Thanks for the suggestion. We included the schematic representation of currents in fig. 1 and improved the description of the circulation patterns.**

L136-137: SACS – Probably T < 13°C? And according to the profile this is below 150 m and not in the first 150 m.

**R. Thanks for the suggestion, we changed to "reaching up to ~150 m depth".**

L141: Why is salinity higher on the shelf in autumn? One would suspect that higher runoff would lower the salinity. Is the influence of rivers limited only to a very narrow belt?

**R. With the higher temperature during autumn, the evaporation/precipitation budget is higher, increasing the overall salinity (Assunção et al. 2020). We improved the text to make it clearer.**

L144-145: The fluorescence indicates river influence over the shelf in autumn. How does this fit with higher salinity?

**R. See our response to the previous comment.**

L159: remove "also"

**R. Thanks for the suggestion. We removed it.**

L160: % of the total catch? By mass or number of individuals?

**R. By number of individuals.100 m$^{-3}$. We improved the text.**

Section 3.3

I find this section a bit confusing and it would need a clearer structure. It switches from hydromedusae to siphonophores and back. The precision of many values is overstated by using too many digits (e.g. 1067.6 ind. 100 mâ• »³). Only the number of reliable digits should be used. I'd suggest to structure the section more clearly and follow the same pattern, otherwise the reader quickly gets lost between species, abundances, locations and seasons.

**R. Thanks for your concern. We reordered the section. About the second part, we are not sure about what you refer as "too many digits". All values were presented with 1 decimal digit.**

L241: There is no figure 8b.

**R. Thanks you, it should be 2b.**

L253: "spreads" – "spread"

**R. Corrected.**

L255-258: The reduction of the current intensity in autumn is not so clear. You discuss the depth and speed of the current and sometimes conclude that the surface current was of similar strength. Fig. 2 would lead me to that conclusion as well. A plot of satellite derived geostrophic currents could be very helpful here.

**R. Thanks for the suggestion. These patterns of NBUC depth and intensity were discussed in details in Dossa et al. 2021, where the same ADCP dataset was used and compared with geostrophic currents.**

L287: How about S. chuni?

**R. Thanks for noticing. We included the species in the list.**

L304-305:  L. meteori is not shown in Fig. 5.  Are these species considered oceanic or coastal?

**R. We included only the more abundant species in Fig. 5. There is not much information about ecological requirements of this species in literature. During our spring cruise, it occurred exclusively in the open ocean, particularly over the slope.**

L306: The second spring is probably autumn.

**R. Thanks you, we corrected it.**

L315-316: I would say that your conclusions are valid for the Brazilian shelf only, since they were not tested elsewhere.

**R. Now we referred to our study area and only suggest that the pattern may be similar in other WBCS.**

L319-321: This is a bold claim given the presented results. It kind of makes sense, but I am sure you have enough data to better support your claim.

**R. Thanks for the concern. We believe that this conclusion was supported by our data and discussed in details in lines 337-349.**

Figure 2: The arrows are way too small and there is too many of them.

**R. Thanks for the suggestion. We improved he figure to make it more readable.**

Figure 3g: There is a huge upper limit of fluorescence in autumn. It would be interesting to see which stations contribute to that. Something like fig 7c but for chlorophyll would help and maybe support your third hypothesis. Maybe a plot of satellite chlorophyll concentration would be useful as well.

**R. Thanks for the suggestion. We included distribution maps of surface temperature, salinity and fluorescence in supplementary material (Fig. S1)**

---

## Author Response (AR2)

Dear Prof. Dr. Mario Hoppema,

Please find the corrected MS of our study "Planktonic cnidarian responses to contrasting thermohaline and circulation seasonal scenarios in a tropical western boundary current system".

We sincerely appreciate the useful insights provided by you and Dr. Martin Vodopivec to improve our study. As detailed in the rebuttal letter below (responses in bold) we took into account all corrections and suggestions to prepare the revised version of our MS.

We hope that you will find that the revised manuscript adequately addressed all comments and that it now is suitable for publication in Ocean Science.

We thank you very much, in advance, for the attention you will grant to our re-submission.

**Editor coments:**

L54-55 „ However, the effects of the variability in western boundary currents intensity, intrusions of tropical water and river runoff over the continental shelf in zooplankton distribution and abundance are to be tested." Do you mean: "However, if the effects …"

**R. No, we mean the effects are "still" to be tested. We corrected it.**

L56 … westward flowing, central South Equatorial Current … (delete: of the)

**R. We corrected it.**

L58 … shallower mixed-layer leads to an increase of the nutrient input …

**R. We corrected it.**

L60 delete: relatively

**R. We deleted it.**

L60-61 I do not understand the second part of this sentence ("and it was for 60 instance related with increased production of zooplankton"), and how it relates to the first part. Please clarify.

**R. We changed to "and it was also related with increased production of zooplankton, though bottom-up control effect".**

Section 2.1 Please specify the salinity values you use in the paper: practical salinity or absolute salinity?

**R. We specified it was practical salinity.**

L89 for temperature use: 0.001 °C instead of 10-3

**R. We corrected it.**

L116 through instead of though

**R. We corrected it.**

L156 data during (typo)

**R. We corrected it.**

L319 "The oceanward flow free up space for the spread of coastal waters over the continental shelf …" It is not quite clear what is meant here. Please rephrase.

**R. We changed to "The oceanward flow opens space over the continental shelf for the spread of coastal water, which also favors the uplift of deeper water masses".**

L338 "the cline between …" What do you mean with cline? Nutricline, thermocline? Please modify.

**R. We changed to "thermocline".**

L344-345 "Anywise, the increase in primary production due changes in mixed layer generally reflects in coupled increase in secondary production of zooplankton" This sentence is not clear. Please rephrase.

**R. We changed to "Anywise, the increase in primary production related to changes in mixed-layer depth typically reflects in increased production in the upper trophic chain in zooplankton, reflecting a bottom-up control".**

L361 delete: In conclusion,

**R. We deleted it.**

5. Conclusions: I would expect some word on the differences between spring and autumn.

**R. We more specifically refer to spring and autumn differences.**

Figure 1: Add a) and b) to the figure panels. Also: Please add all geographic names used in the text to the figure, for example, Pernambuco Plateau.

**R. We included it.**

Figure 2: Please explain what the dashed lines mean. Also: the current vectors are relatively small and it is hard to discern them. Would it be possible to make them clearer?

**R. We explained the dashed lines are the boundaries between WBCS, transition zone and SECS. We reduced the number of arrows and made them thicker.**

Figure 4: Please mention what the star in the different panels indicates

**R. We included it in the caption.**

Figure 5: Please mention what the star in the different panels indicates. Also, give the unit on the y-axis.

**R. We included both in the caption.**

Caption Figure 6: … of the water column in … (add: column)

**R. We included it.**

References

L406-409 What kind of publication is this? Report? Any more info on it?

**R. It is a software manual (reference for the analysis).**

L414 Clarke, K. R. and Gorley, R. N.: PRIMER 6 + PERMANOVA, 2006 What kind of publication is this? Any more info on it?

**R. It is a software. We did not find how to cite it in the guidelines.**

L423-424 Volume and pages should be: Arch. Fish. Mar. Res. 47(2/3), 1999, 5–24

**R. We corrected it.**

L484 Pages?

**R. We included it.**

L495 QGIS Development Team: QGIS Geographic Information System, 2022. What kind of publication is this? Any more info on it? Link? Access date?

**R. It is a software. We did not find how to cite it in the guidelines.**

L504 Volume and pages?

**R. We included it.**

Supplementary material: Please take all figures and tables together in one file and describe (give them captions) and explain. Please see the author instructions for Supplementary Material.

**R. We did it except for Supplementary table A2, which is a large spreadsheet for data sharing.**

**Reviewer comments:**

The authors have mostly modified the paper according to the comments and in my opinion it is now markedly easier to follow their reasoning.

**R. Thanks.**

I would also suggest moving the Figure S1 into the main body of the paper. I find it very informative.

**R. We believe the manuscript is already quite long and including Fig. S1 in the main text would make it even longer. However, if mandatory, we may include it for the final version.**

However, some things should still be corrected:

Abstract: The results of the study are valid for the western boundary system of tropical South Atlantic and not necessarily for all WBCS. This is nicely stated in the Conclusions section, but should be corrected in the abstract as well.

**R. We changed it.**

Figure 2: Adding the current speed to the plots has significantly improved their clarity, but the vectors are still to small to be visible. Especially on prints. I suggest reducing the number of arrows and increasing their thickness.

**R. Thanks for the suggestion. We did it.**

The number of digits/figures in presented values is still too large and makes a false impression of accuracy. Here is a short document about precision and suitable number of figures: https://www2.chem21labs.com/labfiles/jhu_significant_figures.pdf – see point no. 9!

Also: https://faraday.physics.utoronto.ca/PVB/Harrison/ErrorAnalysis/SignificantFigures.html

This is one of the most basic principles in statistics! There are several instances throughout the manuscript and especially in the section 3.3 (but also e.g. Table S1 etc.) where the number of digits is much larger than the accuracy of the result would permit.

**R. Along the whole manuscript (including section 3.3 we used 1 decimal digit for means and SD. In table S1 values were indeed with two decimal digits, we changed it in this new version.**